# Effects of Resistance Training on Motor and Cognitive Function in Older Adults with Alzheimer’s Disease: A Systematic Review

**DOI:** 10.3390/healthcare13233079

**Published:** 2025-11-26

**Authors:** Maria Fernanda Serna-Orozco, Stefania Pitto-Bedoya, Jhoan Sebastián Salazar-Goyes, Sebastián Figueroa-Zúñiga, Luisa María Martínez-Muñoz, Jennifer Jaramillo-Losada

**Affiliations:** 1Research Group Biomateriales y Biotecnología (BEO), Universidad Santiago de Cali, Cali 760035, Colombia; maria.serna05@usc.edu.co; 2Physiotherapy Program, Faculty of Health, Universidad Santiago de Cali, Cali 760035, Colombia; stefaniapitto02@gmail.com (S.P.-B.); salazarjhonsebas@gmail.com (J.S.S.-G.); sebas_15-16@hotmail.com (S.F.-Z.); 3Programa de Fisioterapia, Grupo de Investigación Ejercicio y Salud Cardiopulmonar (GIESC), Universidad del Valle, Cali 760035, Colombia; luisa.martinez.munoz@correounivalle.edu.co; 4Physiotherapy Program, Faculty of Health, Salud y Movimiento Grupo de Investigación, Universidad Santiago de Cali, Cali 760035, Colombia

**Keywords:** Alzheimer’s disease, motor function, cognitive function, muscle strength, physical exercise

## Abstract

**Highlights:**

**What are the main findings?**
Resistance exercise significantly improves muscle strength, motor performance, and functional capacity in older adults with Alzheimer’s disease.Resistance exercise programs lasting at least 12 weeks, performed three times per week at moderate intensities (50–70% of 1RM), appear to represent a safe and potentially neuroprotective intervention for individuals with Alzheimer’s disease.

**What are the implications of the main findings?**
Incorporating resistance exercise into clinical care can help maintain independence and reduce caregiver burden in Alzheimer’s patients.Targeted physical training should be considered a key component of therapeutic strategies for managing Alzheimer’s-related physical decline.

**Abstract:**

**Background/Objectives**: This systematic review aimed to determine the effects of resistance training on cognitive and motor function in older adults diagnosed with Alzheimer’s disease (AD). **Methods**: The review followed PRISMA guidelines. A comprehensive search strategy was applied across MEDLINE (OVID), SCOPUS, Web of Science, LILACS, and the Cochrane Central Register of Controlled Trials (CENTRAL). The included studies were experimental, quasi-experimental, cohort, case–control, and cross-sectional designs. Exclusion criteria included studies in animals, pediatric populations, individuals with other types of dementia, Down syndrome, or other neurodegenerative diseases. **Conclusions**: Resistance training appears to exert beneficial effects on both motor and cognitive functions in older adults with AD. However, the development of standardized, individualized exercise protocols is essential to optimize therapeutic outcomes

## 1. Introduction

Alzheimer’s disease is a progressive neurodegenerative disorder and the most common form of dementia worldwide, accounting for approximately 60% to 80% of all cases among older adults. It currently affects over 55 million people globally [1]. AD is marked by pathological changes in the brain that lead to progressive neurodegeneration and decline in cognitive function, particularly memory. These changes are strongly associated with the abnormal accumulation of β-amyloid plaques and hyperphosphorylated tau protein1, which activate central nervous system immune cells and trigger an inflammatory response that accelerates neuronal damage [2]. Over time, this progression results in total loss of cognitive function and complete dependence on caregivers, as patients lose the ability to perform basic activities of daily living [3].

Physical exercise has been shown to modulate the immune response and inflammatory processes by influencing endogenous markers within the inflammatory cascade, and evidence suggests that physical activity could mitigate the detrimental effects of aging on cognition [4]. In the context of AD, regular moderate-intensity physical activity has been shown in both animal models and human studies to inhibit microglial activation and improve disease pathogenesis by enhancing the brain’s redox state [5], promoting the release of neurotrophic factors, and modulating the molecular mechanisms involved in Aβ production [6].

Among various types of physical activity, resistance training has been found effective in improving skeletal muscle strength, power, and endurance [7]. It also contributes to increased muscle mass and bone mineral density, while inducing neural adaptations, improving cardiometabolic health [8], and reducing symptoms of depression and anxiety [9,10]. Resistance training has been associated with decreased risk of chronic low-grade inflammation [11] and sustained resistance training appears to lower pro-inflammatory cytokine levels both at rest and in response to exercise [12]; increase the secretion of brain-derived neurotrophic factors (BDNF) [13]; and, in animal models of Alzheimer’s disease, has demonstrated the ability to promote Aβ clearance [14] and reduce both the volume and number of Aβ plaques and intracellular neurofibrillary tangles in the brain [6].

When combined with aerobic exercise, resistance training has shown significant benefits for both physical and cognitive functions, as well as improvements in activities of daily living [15]. It has also been proposed as a potentially effective strategy for preventing and treating cognitive decline [16]. Evidence suggests that exercise-induced cognitive improvements may be linked to reduced white matter degradation and favorable hemodynamic changes in various brain regions [17]. However, despite these findings, muscle-strengthening exercises remain underutilized in public health strategies for chronic disease prevention. While the effects of aerobic exercise on motor and cognitive function have been extensively studied, research on the impact of resistance training remains limited. This systematic review aims to identify and synthesize available clinical evidence on the benefits of resistance training for cognitive and motor function in individuals with Alzheimer’s disease.

## 2. Materials and Methods

This systematic review was conducted in accordance with Cochrane Collaboration recommendations and reported following the PRISMA 2020 statement [18]. The review was prospectively registered in PROSPERO (ID: CRD420251144448); URL: https://www.crd.york.ac.uk/PROSPERO/view/CRD420251144448 (accessed on 1 October 2025).

### 2.1. Eligibility Criteria

We included experimental, quasi-experimental, cohort, case–control, and cross-sectional studies that specifically evaluated the effects of exclusive resistance training interventions in adults diagnosed with Alzheimer’s disease; for this study, resistance training is defined as a structured program of exercises in which participants perform movements against an external resistance (such as free weights, machines, resistance bands, or body weight) with the primary aim of improving muscle strength, endurance, and functional capacity [19]. The primary outcome was motor function, assessed through various measures such as the 6 min walk test, handgrip dynamometry, but not limited to these evaluations. The secondary outcome was cognitive function, measured with instruments such as the Mini-Mental State Examination (MMSE) and the Montreal Cognitive Assessment (MoCA), among others, without restricting the type of cognitive assessment considered. No limits were imposed regarding language or publication date; articles published from inception through April 2025 were considered eligible. Studies were excluded if they involved animals, pediatric populations, individuals with comorbidities, other types of dementia, Down syndrome, or other neurodegenerative disorders.

### 2.2. Information Sources

We used Medical Subject Headings (MeSH), Health Sciences Descriptors (DeCS), and free-text terms related to the topic. The databases searched included MEDLINE (OVID), Web of Science, Scopus, LILACS, and the Cochrane Central Register of Controlled Trials (CENTRAL). To ensure comprehensive coverage, we also reviewed reference lists of relevant studies and consulted grey literature sources such as conference proceedings, thesis databases, Google Scholar, and ClinicalTrials.gov.

### 2.3. Search Strategy

A comprehensive search strategy was developed to retrieve relevant studies on the Effects of Resistance Training on Motor and Cognitive Function in Older Adults with Alzheimer’s Disease. The search employed a series of keyword combinations aligned with the specific objective of the study.

The keywords included combinations tailored to each database:Medline: Alzheimer Disease OR “Alzheimer Dementia” OR senile dementia AND Resistance Training OR Muscle Strength OR strength training OR endurance training OR muscle strengthening exercise OR muscle exercise AND Cognitive function OR Motor Activity OR cognition OR cognitive function OR mental function OR motor function OR motor activity OR cognitive ability OR cognitive processes OR cognitive abilities.Scopus: (TITLE-ABS-KEY (“Alzheimer Disease” OR “Alzheimer Dementia” OR senile dementia AND TITLE-ABS-KEY (“Muscle Strength”) OR Resistance Training OR strength training OR endurance training OR muscle strengthening exercise OR muscle exercise ABS (“Cognitive function”) OR ALL (“Motor Activity”)) OR cognition OR cognitive function OR mental function OR motor function OR cognitive ability OR cognitive processes OR cognitive abilities.Web of Science: ((TI = (Alzheimer Disease)) OR “Alzheimer Dementia” OR senile dementia AND AB= (Muscle Strength)) OR Resistance Training OR strength training OR endurance training OR muscle strengthening exercise OR muscle exercise OR AB = (Cognitive function)) OR cognition OR cognitive function OR mental function OR motor function OR cognitive ability OR cognitive processes OR cognitive abilities.Lilacs: (Ti:(Alzheimer Disease)) OR “Alzheimer Dementia” OR senile dementia AND (ab: (Muscle Strength)) OR Resistance Training OR strength training OR endurance training OR muscle strengthening exercise OR muscle exercise (ab:(Resistance Training)) AND (mh: (clinical trial)) OR (tw: (double-blind)) OR (tw: (clinical experiment))) OR cognition OR cognitive function OR mental function OR motor function OR cognitive ability OR cognitive processes OR cognitive abilities.CENTRAL: Alzheimer Disease in Title Abstract Keyword OR “Alzheimer Dementia” OR senile dementia AND Motor Activity in Title Abstract Keyword OR Resistance Training OR strength training OR endurance training OR muscle strengthening exercise OR muscle exercise AND Cognitive function in Title Abstract Keyword) OR cognition OR cognitive function OR mental function OR motor function OR cognitive ability OR cognitive processes OR cognitive abilities.

For grey literature searches, the following combination was used: Alzheimer Disease OR “Alzheimer Dementia” OR senile dementia AND Resistance Training OR Resistance Training OR strength training OR endurance training OR muscle strengthening exercise OR muscle exercise AND Cognitive Dysfunction OR Cognition AND No intervention) OR cognition OR cognitive function OR mental function OR motor function OR cognitive ability OR cognitive processes OR cognitive abilities.

### 2.4. Data Collection

Two reviewers independently screened each reference by title and abstract. Full texts of potentially relevant studies were then reviewed to apply inclusion and exclusion criteria and extract data. Discrepancies were resolved through consensus, and if necessary, a third reviewer was consulted.

Two trained reviewers independently used a standardized form to extract the following data: title, authors, study design, language, publication year, objectives, assessed variables, characteristics of control and experimental groups, type, duration, and frequency of intervention, exercise protocol and modality, participant age and sex, and primary outcomes.

### 2.5. Risk of Bias

Risk of bias was assessed independently by two reviewers. Randomized controlled trials were evaluated using the Cochrane Risk of Bias 2.0 (RoB 2) tool, which examines five domains: (1) bias arising from the randomization process, (2) bias due to deviations from intended interventions, (3) bias due to missing outcome data, (4) bias in measurement of the outcome, and (5) bias in selection of the reported result. Non-randomized studies were evaluated using the ROBINS-I tool, which assesses bias across seven domains, including confounding, selection of participants, classification of interventions, deviations from intended intervention, missing data, measurement of outcomes, and selection of reported results.

For each domain, studies were classified as having low, some concerns, or high risk of bias (RoB 2), or as low, moderate, serious, or critical risk (ROBINS-I). Discrepancies were resolved through consensus. All visual summaries, including traffic-light plots and domain-level bar charts, were generated using the robvis application [20], URL: https://mcguinlu-github-io.translate.goog/robvis/?_x_tr_sl=en&_x_tr_tl=es&_x_tr_hl=es&_x_tr_pto=tc (accessed on 15 November 2025).

## 3. Results

The initial search yielded 1182 articles. After removing duplicates, 955 titles and abstracts were screened, resulting in the exclusion of 875 records due to being systematic reviews, editorials, intervention protocols, or unrelated to the topic of interest. Subsequently, 80 full-text articles were reviewed, of which 73 did not meet the inclusion criteria. Ultimately, 7 studies were included in the qualitative synthesis (Figure 1) (see Table A1 in Appendix A).

### 3.1. Characteristics of Excluded Studies

Articles were excluded for not addressing the intervention or outcomes of interest. Additionally, studies conducted in animals, in populations with conditions other than Alzheimer’s-type dementia, in children, individuals with Down syndrome, or those with other neurodegenerative diseases were excluded.

### 3.2. Characteristics of Included Studies

Seven studies published between 2012 and 2025 were included in this review. Of these, four were randomized controlled trials [21,22,23,24], and three were non-randomized studies [25,26,27]. Among the studies, one was conducted in Greece [23], one in Spain [24], two in South Korea [21,22], and three in Brazil [25,27].

Four studies assessed the effects of therapeutic and resistance training on cognitive and functional abilities in older adults with AD [21,23,25]. Three focused on muscle strength and fall risk. Sample sizes ranged from 23 [26] to 171 [23] participants. Three studies included both sexes [23,24,25], two [21,22] focused exclusively on women over 65, and two [21,27] did not report the sex distribution. Mean participant age ranged from 74 to 82 years, with women predominating in all samples [22,24].

One study included only an intervention group of AD patients performing resistance exercises [21]. The remaining six compared resistance training with a control group of AD patients who either received no intervention or underwent alternative treatments [21,23,25,26].

Intervention protocols varied. Four studies used weighted resistance exercises targeting major muscle groups [23,24,25,27]. Three others implemented low-impact resistance exercises using elastic bands and body weight [21,22,26]. The duration of the interventions was five months in one study [26], 12 weeks in four studies [21,22,23,24], and 16 weeks in two studies [25,27]. Most studies prescribed three sessions per week lasting 30–60 min [21,25,26,27], but one study used five weekly 30 min sessions [22] (see Table A2 and Table A3 in Appendix A).

### 3.3. Risk of Bias Assessment

Among the four RCTs, two studies [23,24] were judged to have an overall low risk of bias, with low risk across most domains and only minor concerns related to outcome measurement. The remaining two trials [21,22] were rated as having some concerns, primarily due to issues in the randomization process, deviations from intended interventions, and uncertainty regarding blinding of outcome assessors (see Figure 2 and Figure 3).

In contrast, all three non-randomized studies evaluated with ROBINS-I demonstrated an overall serious risk of bias [25,26,27]. The most critical limitation across these studies was the serious risk of confounding, as none controlled for baseline differences or other potential confounders. Additional moderate risks were identified in participant selection, deviations from intended interventions, outcome measurement, and selective reporting. Two studies showed a serious risk due to missing outcome data, further reducing confidence in their findings (see Figure 4 and Figure 5).

### 3.4. Cognitive Function Evaluation

Five studies assessed cognitive function using the MMSE [21,22,25,26,27]. Although patients improved physically, no significant differences in cognitive function were observed between the intervention and control groups. This may be due to cognitive assessment being performed only at baseline, with no follow-up or post-intervention evaluations.

One study using the Trail Making Test and ACE-R reported global cognitive improvement in the resistance training group, including attention, orientation, memory, verbal fluency, language, and visuospatial abilities. This group also showed greater independence in daily activities [23]. Another study using the clock drawing and verbal fluency tests found no significant cognitive changes attributable to resistance training [25].

### 3.5. Motor Function Evaluation

All the included studies assessed motor function. One used an 800 m walk test, showing significant endurance gains in the intervention group [27]. Two [21,24] assessed handgrip strength using dynamometry with three 30 min sessions per week. Another evaluated [27] lower limb strength through the chair rise test, finding improvements in agility, balance, flexibility, and strength, as well as reduced chair-rise time.

Two studies used isometric maximal voluntary contraction (MVC) to assess strength. One found improved MVC in the hip and knee, while the other extended the assessment to include the shoulder and elbow. Both reported functional improvements [21,27].

Finally, one study used the Short Physical Performance Battery (SPPB) to assess fall risk and strength, reporting enhanced quality of life, reduced fall risk, and improved grip strength in the resistance group versus controls [24].

### 3.6. Synthesis Approach

Estimates could not be quantitatively pooled due to substantial heterogeneity in intervention types (elastic bands, weight training, bodyweight, and variable intensities), outcome measures, and sample sizes, as well as the inclusion of non-RCT designs. Under these conditions, a pooled meta-analytic effect would have resulted in very low certainty and extremely high heterogeneity (I^2^ > 75%). Therefore, we conducted a structured narrative synthesis using vote counting based on the direction of effect (see Table A4 in Appendix A) and assessed the certainty of evidence for each prespecified outcome using the GRADE framework (see Table A5 in Appendix A). All seven studies were included in the vote-counting analysis. Overall, the direction of effect favored resistance or multimodal exercise interventions in adults with mild to moderate AD. For motor function, vote counting indicated a clear beneficial effect, with positive findings across nearly all studies evaluating this domain (see Table A6 in Appendix A). For cognitive function, the direction of effect showed only modest improvements; the evidence was limited by heterogeneity in cognitive assessments, absence of follow-up measures, and small sample sizes. For activities of daily living, although only a small number of studies contributed data, all demonstrated a consistently positive direction of effect.

The overall certainty of evidence ranged from moderate to very low across outcomes. Motor function had the highest certainty (graded as moderate); despite some methodological limitations, effects on gait speed, muscle strength, and physical performance were consistent across studies. In contrast, the certainty for cognitive outcomes was rated as very low due to methodological limitations, variability in assessment tools, and small, imprecise effects. Evidence for activities of daily living and risk of falls was also graded as very low, primarily due to the small number of studies and inconsistent findings. Finally, no study reported adverse events, and the absence of safety data introduced uncertainty regarding potential harms, resulting in downgrading for indirectness and suspected reporting bias.

## 4. Discussion

This systematic review demonstrates that resistance training has a positive impact on individuals with Alzheimer’s disease, particularly in enhancing motor function, with notable improvements in muscle strength and cardiorespiratory endurance. Cognitive benefits were also observed, although the evidence was more limited and variable.

Motor function showed the most consistent improvements across the included studies. Significant gains in physical performance were reported, as assessed through cardiorespiratory and strength-related outcomes. These findings are consistent with previous research indicating that resistance training improves muscular strength and endurance in healthy older adults, even with interventions of fewer than 24 weeks [28]. In this review, the included studies reported benefits in both cardiorespiratory capacity and muscular strength in individuals with AD, even with 12-week programs performed at a minimum frequency of three sessions per week [29]. Systematic reviews and experimental studies support that structured resistance exercise can contribute to improvements in cognitive function, neuroplasticity, systemic inflammation, functional status, and physical capacity in people with AD. Evidence indicates that programs lasting 8–12 weeks or longer are the ones that consistently show effects on cognition, physical function, and biomarkers (BDNF, IGF-1). Interventions of 12 weeks with three sessions per week are commonly used as a standard in randomized controlled trials (RCTs) and have shown cognitive and strength improvements [30,31,32,33,34,35]. Evidence also suggests that moderate intensities can produce gains in strength and metabolic and neurotrophic changes without the higher risk of injury associated with very high intensities (>80% 1RM). Additionally, when direct 1RM testing is not feasible, submaximal tests or RPE (rating of perceived exertion) are recommended [36,37].

In animal models of Alzheimer’s disease, physical exercise—including resistance training and multimodal programs—has been shown to reduce Aβ levels and accumulation, as well as increase the activity or expression of degrading enzymes such as neprilysin and IDE. Taken together, these preclinical findings support the biological plausibility of exercise-related benefits, although their translation to humans still requires stronger evidence [38,39].

Loss of muscle strength and muscle atrophy are common changes associated with aging and have been linked to an increased risk of developing [40] AD. Reduced muscle strength contributes to functional decline, limiting the ability to perform daily activities and maintain an active lifestyle. In this context, resistance training has been proposed as an effective strategy to increase muscle mass, strength, balance, and functional capacity [41]. In a study conducted by Rolland et al. [42], the implementation of resistance exercises with light weights and elastic bands in individuals with AD produced significant gains in muscle strength, as evidenced by improved performance on the six-minute walk test. These results demonstrated that resistance training contributed to improvements in agility, strength, and endurance in older adults, leading to increased walking speed. These findings align with the results of the present review, in which physical endurance improved in the intervention groups. The included studies reported significant increases in lower-limb strength, assessed through maximal voluntary contraction, as well as improvements in handgrip strength measured with dynamometry. Functional performance also improved, with better execution of tasks such as standing from a chair, walking short distances, but the evidence regarding fall-risk reduction was limited, as only one study reported positive effects in this domain. Collectively, these results suggest that resistance training not only has a positive impact on muscle strength but may also enhance functional independence in individuals with AD, supporting greater autonomy in daily living activities.

Previous research has shown that resistance exercise significantly improves cognitive function in older adults with cognitive frailty, particularly in composite cognitive scores and cognitive screening tests [43]. In the studies included in this review, just one article showed some cognitive improvement observed in participants with AD when assessed with the Addenbrooke’s Cognitive Examination—Revised, the Trail Making Test, and the Mini-Mental State Examination. However, these findings were not consistent across studies; most studies using the MMSE did not report follow-up cognitive assessment, as the MMSE was applied only at baseline to classify cognitive impairment, without follow-up assessments during or after intervention. Therefore, rather than concluding if strength training results in cognitive improvement, the evidence from the included studies is insufficient to infer meaningful post-intervention changes in cognition.

It is important to note that the progression of AD itself may limit regular participation in physical activity, thereby reducing the potential for cognitive benefits. Nevertheless, resistance exercise has shown positive effects not only on physical health but also on cognitive function. Such training promotes the release of neurotrophic factors, such as brain-derived neurotrophic factor, which plays a key role in neuronal plasticity, neurogenesis, and the survival of hippocampal neurons [44]. Furthermore, resistance training improves cerebral blood flow and oxygenation, facilitating neuronal regeneration and helping to mitigate the cognitive decline characteristic of AD. It has also been shown to reduce neuroinflammation and to enhance executive functions by promoting the integration of memory with motor control, thus potentially slowing cognitive decline in older adults.

Sepúlveda-Lara et al. indicate that resistance training reduces the levels of pro-inflammatory cytokines such as tumor necrosis factor alpha (TNF-α) and interleukin-6 (IL-6), and increases the levels of anti-inflammatory cytokines such as IL-10. This, in turn, decreases intracellular neurofibrillary tangles (NFTs) in the brains of mice with Alzheimer’s disease. Resistance exercise could represent an alternative therapy for preventing and treating Alzheimer’s disease, as it has also been shown to have a significant effect on IGF-1 levels—being most effective with three weekly sessions [35]. Additionally, evidence indicates that resistance training enhances neuroplasticity through neurotrophic signaling (BDNF, IGF-1, and VEGF), neuroendocrine regulation, epigenetic modifications, and the optimization of metabolic pathways. In animal models, it also reduces Aβ burden by increasing the expression of Aβ-degrading enzymes such as neprilysin and insulin-degrading enzyme (IDE) [45,46,47].

Despite the documented benefits, the inclusion of resistance exercise in therapeutic strategies for individuals with AD remains limited. Most interventions have focused predominantly on aerobic exercise [48,49], even though evidence indicates that strength training could complement and enhance the effects of other treatment modalities. Therefore, future research should prioritize identifying the most effective resistance training protocols, with clearly defined parameters for intensity, duration, and frequency, in order to maximize clinical benefits in this population.

## 5. Limitations of the Study

Although the results are generally favorable, this review presents several limitations related to the studies that investigated the effects of resistance exercise on cognitive and motor function in older adults with Alzheimer’s disease. One major limitation is the lack of standardization across exercise protocols, which complicates direct comparisons between studies. Marked variations in intensity, duration, and types of resistance exercises were identified—factors that can significantly influence the magnitude of the reported effects. These discrepancies highlight the need for more uniform and clearly defined training parameters in future research to enhance comparability and strengthen the reliability of findings. Another important limitation is the insufficient follow-up of cognitive outcomes. In several studies, cognitive function was assessed only at baseline (e.g., using the MMSE), without subsequent post-intervention evaluations. This lack of longitudinal assessment restricts the ability to determine the true impact of resistance training on cognitive decline. Furthermore, methodological issues such as inadequate blinding procedures and small sample sizes were common across studies, reducing both the internal and external validity of the results and limiting the generalizability of the conclusions. The limited representation and analysis of subgroups also represent a relevant constraint, as potential differences related to gender, disease severity, or comorbidities were not explored—factors that are essential for the development of personalized training protocols. For instance, Papatsimpas et al. (2023) [23] reported no significant differences among participant groups in variables such as age, BMI, or years of education; however, the study did not examine variations associated with gender, Alzheimer’s severity, or comorbidities, a limitation also observed in Ahn et al. (2015) [26]. Similarly, Yun et al. (2021) [22] and Chang et al. (2020) [21] reported no significant demographic differences between groups, although Chang and colleagues noted that resistance training reduced depressive symptoms in older adults with Alzheimer’s disease. Likewise, Garuffi et al. (2013) [27] and Vital et al. (2012) [25] did not include subgroup analyses based on gender, disease severity, or comorbidities, whereas Cámara-Calmaestra et al. (2025) [24] found that resistance training reduced neuropsychiatric symptoms after three months of intervention. Overall, the combination of methodological limitations, insufficient standardization of exercise protocols, and heterogeneity in both interventions and outcome measures reduces confidence in the available evidence and constrains the generalizability of the findings. These issues underscore the need for more rigorous and harmonized research designs in future studies.

## 6. Clinical Implications

The findings of this review support the potential role of resistance exercise as a therapeutic strategy in the management of older adults with AD. From a clinical perspective, resistance training programs may help to preserve functional capacity by targeting two key components of disease-related decline: loss of muscle strength and reduced motor performance. By improving physical endurance, balance, and muscular strength, resistance training enhances mobility and, consequently, can improve patients’ quality of life.

However, the evidence must be interpreted with caution. Variability in exercise protocols, small sample sizes, and methodological limitations in several studies reduce the certainty of the findings. Additionally, heterogeneity in outcome measures and interventions limits direct comparisons and the generalizability of results to all patients with AD.

## 7. Conclusions

Resistance exercise appears to have beneficial effects on motor function, while evidence for cognitive benefits remains limited. However, it is essential to advance toward the development of standardized exercise protocols tailored to the individual needs of this population. Achieving this goal requires a deeper understanding of the physiological mechanisms underlying the benefits of resistance training, enabling a more comprehensive and effective therapeutic approach aimed at optimizing clinical and functional outcomes.

## Figures and Tables

**Figure 1 healthcare-13-03079-f001:**
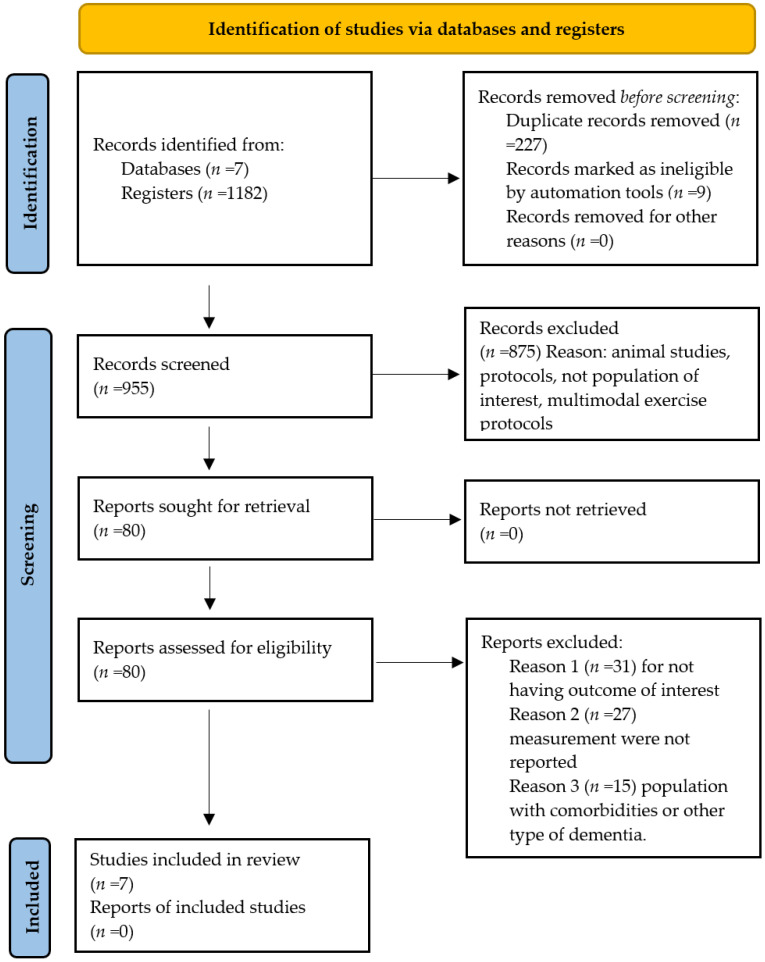
Flowchart of study selection.

**Figure 2 healthcare-13-03079-f002:**
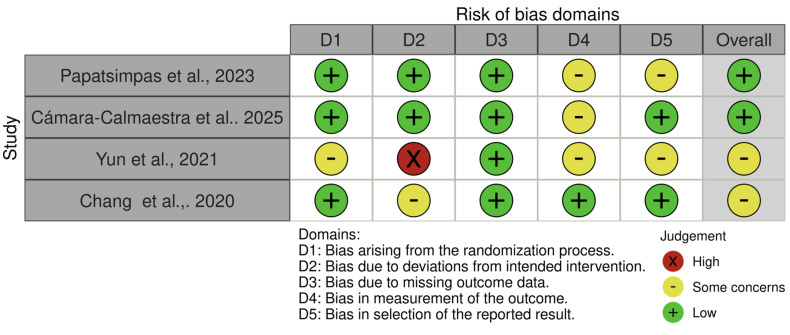
Risk of bias assessment, [21,22,23,24].

**Figure 3 healthcare-13-03079-f003:**
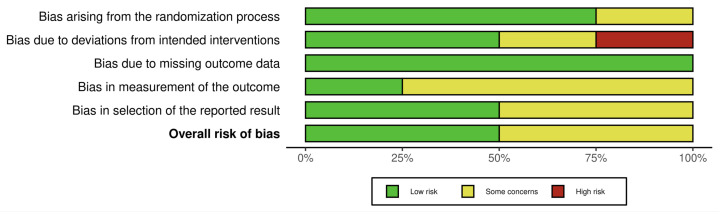
Risk of bias assessment.

**Figure 4 healthcare-13-03079-f004:**
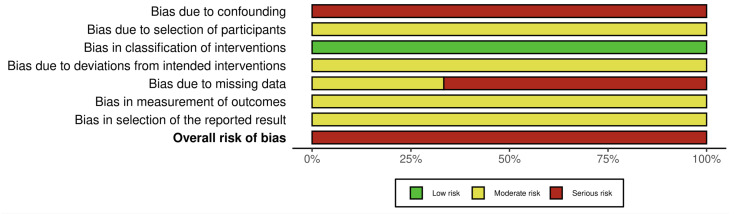
Risk of bias assessment.

**Figure 5 healthcare-13-03079-f005:**
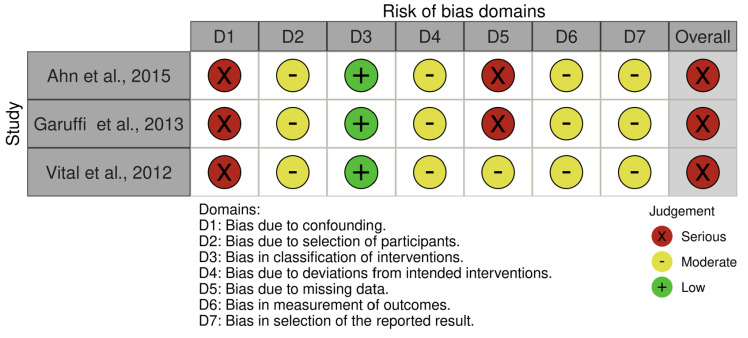
Risk of bias assessment, [25,26,27].

## Data Availability

Data are contained within the article. The original contributions presented in this study are included in the article. Further inquiries can be directed to the corresponding author. We are fully committed to research transparency and are open to sharing the dataset with other researchers for non-commercial and academic purposes, in accordance with ethical considerations. This systematic review is registered in the International Prospective Register of Systematic Reviews (PROSPERO 2025) CRD420251144448. Available from https://www.crd.york.ac.uk/PROSPERO/view/CRD420251144448 (accessed on 1 October 2025).

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
