# Peer review of "Effects of Resistance Training on Motor and Cognitive Function in Older Adults with Alzheimer’s Disease: A Systematic Review"

_healthcare, 2025, doi:10.3390/healthcare13233079_

Round 1

Reviewer 1 Report

Comments and Suggestions for Authors

I sincerely thank the authors for their effort in addressing an important and clinically relevant topic—the effects of resistance training on motor and cognitive function in older adults with Alzheimer’s disease. The research question is pertinent, the scope of the search is broad, and the synthesis highlights promising functional benefits. However, several methodological inconsistencies and reporting limitations substantially affect the robustness and reproducibility of the findings. The manuscript requires significant revisions to improve the alignment between inclusion criteria and included studies, risk-of-bias assessment, data synthesis strategy, and adherence to PRISMA 2020 reporting standards.

1) Overall Recommendation
Major revision.
The review contains several major issues: (i) inconsistencies between stated inclusion criteria (“exclusive resistance training”) and the characteristics of included studies (some multimodal interventions), (ii) contradictions between text and tables in risk-of-bias reporting, (iii) lack of meta-analytical synthesis without formal justification, (iv) incomplete search strategy and absence of a GRADE assessment, and (v) numerous editorial and stylistic inconsistencies.

2) Major Comments (Methodology and Validity)
2.1. PICO and Eligibility Criteria
The inclusion criteria specify “exclusive resistance training interventions,” yet at least one randomized trial combined resistance with aerobic exercise. This contradicts the stated aim and reduces internal validity. The authors should either (a) restrict the analysis to studies using resistance training exclusively or (b) redefine the PICO question and stratify the synthesis between “resistance-only” and “multimodal” interventions.
2.2. Primary and Secondary Outcomes
The manuscript defines motor function as the primary outcome and cognition as secondary. However, several included studies did not report post-intervention cognitive measures (e.g., MMSE listed as “not reported”), yet the discussion interprets cognitive improvement. Such conclusions are not supported by the data and should be reformulated as “insufficient evidence for cognitive change.”
2.3. Risk of Bias Assessment
The tools used (PEDro and MINORS) are acceptable but outdated for current standards. It is strongly recommended to employ RoB 2 for randomized trials and ROBINS-I for non-randomized designs. The text also contains inconsistencies: it states that “six studies scored 9 on PEDro,” whereas the table shows only two studies with a score of 9. Similarly, one study included with a MINORS score of 11 does not meet the threshold (>11) specified in the Methods section. These contradictions must be corrected and explicitly justified.
2.4. Quantitative Synthesis and Certainty of Evidence
When at least two studies assess comparable outcomes (e.g., handgrip strength, MVC, SPPB, gait speed), a meta-analysis should be attempted. The absence of such synthesis must be justified statistically (heterogeneity, lack of comparable metrics, etc.). A random-effects model with SMD or MD and 95% CI is recommended. The authors should also grade the certainty of evidence for each key outcome using the GRADE approach.
2.5. Outcomes on Falls and Quality of Life
Conclusions regarding fall risk and quality of life seem to rely primarily on a single trial. These effects should therefore be described as “limited evidence” rather than generalized. Confidence intervals and effect sizes should be reported whenever available.
2.6. PROSPERO Registration and Transparency
The manuscript includes typographical errors in the PROSPERO code and URL (“PROS-PERO”). Please confirm the registration number, provide the full link, and specify whether the protocol was registered prospectively before data extraction began.

3) Search Strategy and PRISMA Compliance
3.1. Search Strategy Completeness
The last search date is missing, and the Boolean syntax shows redundancy and misaligned terms (e.g., inclusion of “endurance training” for a review focused on strength). The search strategy should be fully reproducible for each database, including search fields, operators, filters, and date of execution. The addition of grey literature sources and ClinicalTrials.gov (with precise filters) is recommended.
3.2. PRISMA Flow Diagram and Study Selection
The flow diagram appears incomplete or formatted as a template. The revised version should include all standard PRISMA 2020 elements and a table of excluded studies with explicit reasons for exclusion.

4) Data Extraction and Intervention Characteristics
A detailed matrix of intervention characteristics is required, including Frequency, Intensity (e.g., %1RM or RPE), Time (minutes per week), Type (machine, free weights, elastic bands), total volume (sets × repetitions × exercises), progression scheme, supervision, and adherence (percentage of sessions completed). For elastic-band protocols, specify how tension or color was standardized. Adverse events should also be systematically reported.

5) Interpretation and Discussion
5.1. Scope of the Conclusions
The manuscript currently states that resistance training improves both motor and cognitive outcomes. Given the limited and heterogeneous cognitive data, this conclusion should be nuanced. The strength of evidence is moderate for motor outcomes (strength, functional performance) and low to very low for cognition, due to small samples, inconsistent measures, and high clinical heterogeneity.
5.2. Mechanistic Rationale
The proposed biological mechanisms (e.g., BDNF modulation, cerebral perfusion) are relevant but should be explicitly linked to the exercise dose used in included studies (intensity, duration, volume). Otherwise, these sections risk overinterpretation.

6) Reporting Quality and Editorial Issues

  • The “Highlights” section should avoid categorical statements such as “RCTs demonstrate...” when only limited evidence exists.
  • Ensure consistent terminology (MVC, MMSE, 1RM, IADL) and standardized abbreviations.
  • The “Patents” section is irrelevant for this type of study and should be removed.
  • The acknowledgment of AI-assisted language editing (ChatGPT) should follow MDPI’s policy on “Use of AI in Manuscript Preparation.”
  • Table and figure numbering must be harmonized.
  • Verify all references for completeness (authors, years, DOI) and ensure consistency with in-text citations.

7) Specific Analytical and Structural Recommendations

  1. Reframe the PICO or stratify analyses (resistance-only vs. multimodal).
  2. Update and clearly document the search strategy, including final search date and complete Boolean syntax.
  3. Reassess risk of bias using RoB 2 and ROBINS-I; correct contradictions with tables.
  4. Conduct meta-analyses when possible for outcomes such as handgrip strength, MVC, SPPB, and gait speed, reporting SMD/MD with 95% CI and heterogeneity (I²).
  5. Perform GRADE assessment for key outcomes and include a Summary of Findings table.
  6. Conduct subgroup analyses (exclusive resistance, duration ≥12 weeks, intensity ≥70% 1RM, home-based vs. supervised).
  7. Rewrite conclusions to reflect graded certainty of evidence.
  8. Remove the “Patents” section and format funding/conflict statements according to journal guidelines.
  9. Perform a complete editorial revision for grammar, abbreviations, table numbering, and consistency.

Author Response

2) Major Comments (Methodology and Validity)

2.1. PICO and Eligibility Criteria The inclusion criteria specify “exclusive resistance training interventions,” yet at least one randomized trial combined resistance with aerobic exercise. This contradicts the stated aim and reduces internal validity. The authors should either (a) restrict the analysis to studies using resistance training exclusively or (b)redefine the PICO question and stratify the synthesis between “resistance-only” and “multimodal” interventions.

Response: All included articles had at least one group receiving only strength training. Results from all groups were reported to allow for comparative analysis. One study combined resistance and aerobic exercise and was structured as a three-group comparison (resistance + aerobic / resistance only / control). 

2.2. Primary and Secondary Outcomes The manuscript defines motor function as the primary outcome and cognition as secondary. However, several included studies did not report post-intervention cognitive measures (e.g., MMSE listed as “not reported”), yet the discussion interprets cognitive improvement. Such conclusions are not supported by the data and should be reformulated as “insufficient evidence for cognitive change.”

Response: Thank you, we have revised the discussion to clarify that the evidence for cognitive improvement is inconsistent and limited.

2.3. Risk of Bias Assessment The tools used (PEDro and MINORS) are acceptable but outdated for current standards. It is strongly recommended to employ RoB 2 for randomized trials and ROBINS-I for non-randomized designs. The text also contains inconsistencies: it states that “six studies scored 9 on PEDro,” whereas the table shows only two studies witha score of 9. Similarly, one study included with a MINORS score of11 does not meet the threshold (>11) specified in the Methods section. These contradictions must be corrected and explicitlyjustified.

Reponse: Thank you very much for your comments. The requested correction has been made.

2.4. Quantitative Synthesis and Certainty of Evidence When at least two studies assess comparable outcomes (e.g.,handgrip strength, MVC, SPPB, gait speed), a meta-analysis should be attempted. The absence of such synthesis must bejustified statistically (heterogeneity, lack of comparable metrics,etc.). A random-effects model with SMD or MD and 95% CI is recommended. The authors should also grade the certainty of evidence for each key outcome using the GRADE approach.

Reponse: Thank you very much for your comments. The requested correction has been made.

2.5. Outcomes on Falls and Quality of Life Conclusions regarding fall risk and quality of life seem to rely primarily on a single trial. These effects should therefore be described as “limited evidence” rather than generalized. Confidence intervals and effect sizes should be reported whenever available.

Reponse: Thank you very much for your comments. Regarding quality-of-life outcomes, we do not include measures of quality of life in the description. Two articles reported outcomes related to instrumental activities of daily living (IADL), and we decided to include this measure because it reflects functional performance closely associated with mobility in this population. Regarding risk of falls we made the requested correction.

2.6. PROSPERO Registration and Transparency The manuscript includes typographical errors in the PROSPERO code and URL (“PROS-PERO”). Please confirm the registration number, provide the full link, and specify whether the protocol was registered prospectively before data extraction began.

Response: Protocol was registered prospectively under the registration number CRD420251144448 and url: PROSPERO

3) Search Strategy and PRISMA Compliance

3.1. Search Strategy Completeness The last search date is missing, and the Boolean syntax show redundancy and misaligned terms (e.g., inclusion of “endurance training” for a review focused on strength). The search strategy should be fully reproducible for each database, including search fields, operators, filters, and date of execution. The addition of grey literature sources and ClinicalTrials.gov (with precise filters) is recommended.

Response: Thank you very much for your comments. The last search date was included in the text. Grey literature is already in search strategy table and mention inside the text under information sources. We made the suggested corrections in search strategy.

3.2. PRISMA Flow Diagram and Study Selection The flow diagram appears incomplete or formatted as a template. The revised version should include all standard PRISMA 2020 elements and a table of excluded studies with explicit reasons for exclusion.

Response: Thank you very much for your comments. The requested correction has been made.

4) Data Extraction and Intervention Characteristics: A detailed matrix of intervention characteristics is required, including Frequency, Intensity (e.g., %1RM or RPE), Time (minutes per week), Type (machine, free weights, elastic bands), total volume (sets × repetitions × exercises), progression scheme, supervision, and adherence (percentage of sessions completed).For elastic-band protocols, specify how tension or color was standardized. Adverse events should also be systematically reported.

Response: Thank you very much for your comments. We know included a new table with the description of interventions of included studies.

5) Interpretation and Discussion

5.1. Scope of the Conclusions The manuscript currently states that resistance training improves both motor and cognitive outcomes. Given the limited and heterogeneous cognitive data, this conclusion should be nuanced. The strength of evidence is moderate for motor outcomes(strength, functional performance) and low to very low for cognition, due to small samples, inconsistent measures, and high clinical heterogeneity.

Response: Thank you very much for your comments. The requested correction has been made.

5.2. Mechanistic Rationale proposed biological mechanisms (e.g., BDNF modulation,cerebral perfusion) are relevant but should be explicitly linked tothe exercise dose used in included studies (intensity, duration, volume). Otherwise, these sections risk overinterpretation.

Response: Thank you very much for your comments. The requested correction has been made.

6) Reporting Quality and Editorial Issues

The “Highlights” section should avoid categorical statements such as “RCTs demonstrate...” when only limited evidence exists.

Ensure consistent terminology (MVC, MMSE, 1RM, IADL) and standardized abbreviations.

The “Patents” section is irrelevant for this type of study and shouldbe removed.

The acknowledgment of AI-assisted language editing (ChatGPT)should follow MDPI’s policy on “Use of AI in Manuscript Preparation.”

Table and figure numbering must be harmonized.

Verify all references for completeness (authors, years, DOI) and ensure consistency with in-text citations.

Response: Thank you very much for your comments. The requested correction has been made.

7) Specific Analytical and Structural Recommendations

Reframe the PICO or stratify analyses (resistance-only vs. Multimodal).

Update and clearly document the search strategy, including final search date and complete Boolean syntax.

Reassess risk of bias using RoB 2 and ROBINS-I; correct contradictions with tables.

Conduct meta-analyses when possible for outcomes such ashandgrip strength, MVC, SPPB, and gait speed, reportingSMD/MD with 95% CI and heterogeneity (I²).

Perform GRADE assessment for key outcomes and include aSummary of Findings table.

Conduct subgroup analyses (exclusive resistance, duration ≥12weeks, intensity ≥70% 1RM, home-based vs. supervised).

Rewrite conclusions to reflect graded certainty of evidence.

Remove the “Patents” section and format funding/conflictstatements according to journal guidelines.

Perform a complete editorial revision for grammar, abbreviations, table numbering, and consistency.

Response: Thank you very much for your comments. The requested correction has been made.

Reviewer 2 Report

Comments and Suggestions for Authors

COMMENT 1: It would be helpful (for purposes of clarity and replication) if the authors provided a PICO informed framework that informed the search strategy.

COMMENT 2: Was the data search and synthesis informed by a structured framework (e.g., the JBI framework)? If so this needs to be specified. If not, what author developed  framework was employed?

COMMENT 3: Please provide a clear operational definition for the term “resistance training” as used in the study

COMMENT 4: Please highlight more clearly in the discussion section what novel insights (if any) emerged from the review and what the implications of such insights may be for practice and future research

Author Response

COMMENT 1: It would be helpful (for purposes of clarity and replication) if the authors provided a PICO informed framework that informed the search strategy.

Response: Thank you very much for your comments. The requested correction has been made and the search strategy in clarify in appendix 1.

COMMENT 2: Was the data search and synthesis informed by a structured framework (e.g., the JBI framework)? If so this needs to be specified. If not, what author developed framework wasemployed?

Response: The data search and synthesis for our review were guided by the Cochrane framework. We followed Cochrane’s structured methodology for systematic reviews, including the development of a predefined protocol, comprehensive database searches, independent screening by two reviewers, standardized data extraction, and structured synthesis of the results. This framework has now been explicitly specified in the Methods section of the manuscript.

COMMENT 3: Please provide a clear operational definition for the term “resistance training” as used in the study

Response:  Thank you very much for your comments. The requested correction has been made.

COMMENT 4: Please highlight more clearly in the discussion section what novel insights (if any) emerged from the review and what the implications of such insights may be for practice and future research

Response: Thank you very much for your comments. The requested correction has been made.

Reviewer 3 Report

Comments and Suggestions for Authors

In the article entitled “Effects of Resistance Training on Motor and Cognitive Function in Older Adults with Alzheimer's Disease: A Systematic Review,” the authors conduct a systematic review of motor and cognitive decline in Alzheimer's disease. The authors provide evidence on a no-pharmacological approach that has an impact on patients' quality of life and functional independence.

There are some important points to address.

  1. Absence of quantitative analysis: Although this is a systematic review, it includes no combined effect estimates or statistical analyses of heterogeneity or publication bias, which limits the strength of the conclusions.
  2. Lack of standardization in exercise protocols: The included studies show great variability in intensity, duration, frequency, and type of resistance applied (free weights, elastic bands, bodyweight exercises), making it difficult to compare results.
  3. Insufficient cognitive assessment: In several studies, cognition was assessed only at baseline (e.g., MMSE), with no subsequent follow-up. This makes it impossible to determine the real effect of training on cognitive decline.
  4. Methodological limitations of the primary studies: Deficiencies in blinding and small sample sizes are acknowledged, which reduces the internal and external validity of the results.
  5. Poor representation of subgroups: Possible differences based on gender, disease severity, or comorbidities are not analyzed, which would be relevant for adapting personalized protocols.

Suggestions.

  1. Include a meta-analysis: If the data allow it, the inclusion of a meta-analysis (e.g., changes in muscle strength or MMSE) would increase the impact and robustness of the manuscript.
  2. Standardization of protocol descriptions: It is recommended to provide an abstract of the key variables (intensity, duration, frequency, volume, and modality) in a standardized comparative table to facilitate interpretation.
  3. Delve deeper into physiological mechanisms: Expanding the discussion on the relationship between resistance, neuroplasticity, and neurogenesis (via BDNF, IGF-1, cytokines) would strengthen the biological basis of the findings.
  4. The manuscript would benefit greatly from improved visual presentation: Incorporate an abstract figure  that synthesizes the effects of resistance training on motor and cognitive domains in AD.
  5. Clarify the role of optimal intensity and duration: Discuss practical recommendations for clinicians or therapists based on available evidence (e.g., ≥12 weeks, 3 sessions/week, 50–70% of 1RM).

6. Adjust the language: A linguistic review is suggested to improve fluency, avoid redundancies, and reinforce terminological accuracy (e.g., distinguish between “resistance training” and “strength exercise”).

Author Response

COMMENT 1: Include a meta-analysis: If the data allow it, the inclusion of ameta-analysis (e.g., changes in muscle strength or MMSE) would increase the impact and robustness of the manuscript.

Response: Thank you very much for your comments. The requested correction has been made.

COMMENT 2: Lack of standardization in exercise protocols: The included studies show great variability in intensity, duration, frequency,and type of resistance applied (free weights, elastic bands,bodyweight exercises), making it difficult to compare results

Response: Thank you very much for your comments. The requested correction has been made.

COMMENT 3: Insufficient cognitive assessment: In several studies, cognition was assessed only at baseline (e.g., MMSE), with nosubsequent follow-up. This makes it impossible to determinethe real effect of training on cognitive decline.

Response: Thank you very much for your comments. The requested correction has been made.

COMMENT 4: Methodological limitations of the primary studies: Deficiencies inblinding and small sample sizes are acknowledged, whichreduces the internal and external validity of the results

Response: Thank you very much for your comments. The requested correction has been made.

COMMENT 5: Poor representation of subgroups: Possible differences basedon gender, disease severity, or comorbidities are not analyzed,which would be relevant for adapting personalized protocols

Response: Thank you very much for your comments. The requested correction has been made.

Suggestions

COMMENT 1: Include a meta-analysis: If the data allow it, the inclusion of ameta-analysis (e.g., changes in muscle strength or MMSE) would increase the impact and robustness of the manuscript

Response: Thank you very much for your comments. The requested correction has been made.

COMMENT 2: Standardization of protocol descriptions: It is recommended toprovide an abstract of the key variables (intensity, duration,frequency, volume, and modality) in a standardized comparativetable to facilitate interpretation.

Response: Thank you very much for your comments. The requested correction has been made

COMMENT 3: Delve deeper into physiological mechanisms: Expanding the discussion on the relationship between resistance, neuroplasticity, and neurogenesis (via BDNF, IGF-1, cytokines) would strengthen the biological basis of the findings.

Response: Thank you very much for your comments. The requested correction has been made

COMMENT 4: The manuscript would benefit greatly from improved visual presentation: Incorporate an abstract figure that synthesizes the effects of resistance training on motor and cognitive domains in AD.

Response: Thank you very much for your comments. The requested correction has been made

COMMENT 5: Clarify the role of optimal intensity and duration: Discuss practical recommendations for clinicians or therapists based on available evidence (e.g., ≥12 weeks, 3 sessions/week, 50–70%of 1RM).

Response: Thank you very much for your comments. The requested correction has been made

COMMENT 6: Adjust the language: A linguistic review is suggested to improvefluency, avoid redundancies, and reinforce terminological accuracy (e.g., distinguish between “resistance training” and “strength exercise”).

Response: Thank you very much for your comments, the language was revised and adjusted.

Round 2

Reviewer 3 Report

Comments and Suggestions for Authors

Thank you to the authors for their replies. With the changes implemented, the manuscript has been substantially improved.